# The Multidimensional Impacts of Inequities for Tāngata Whaikaha Māori (Indigenous Māori with Lived Experience of Disability) in Aotearoa, New Zealand

**DOI:** 10.3390/ijerph192013558

**Published:** 2022-10-19

**Authors:** Tristram Richard Ingham, Bernadette Jones, Meredith Perry, Paula Toko King, Gabrielle Baker, Huhana Hickey, Rangi Pouwhare, Linda Waimarie Nikora

**Affiliations:** 1Department of Medicine, University of Otago, Wellington 6021, New Zealand; 2Centre for Health, Activity and Rehabilitation Research, School of Physiotherapy, University of Otago, Wellington 6021, New Zealand; 3Te Rōpū Rangahau Hauora a Eru Pōmare, University of Otago, Wellington 6021, New Zealand; 4Baker Consulting, Ltd., Wellington 6011, New Zealand; 5Director Pukenga Consultancy Ltd., Auckland 2158, New Zealand; 6Poumanukura Mana Ātea, Whakatane 3192, New Zealand; 7Ngā Pae o te Māramatanga, University of Auckland, Auckland 1010, New Zealand

**Keywords:** disability, equity, inequities, Indigenous Māori, tāngata whaikaha Māori, multidimensional impacts, intersectional disadvantage, racism, ableism, disablism

## Abstract

People with lived experience of disability have poorer health and socioeconomic outcomes than people without it. However, within this population, certain social groups are more likely to experience poorer outcomes due to the impacts of multiple intersecting forms of oppression including colonisation, coloniality and racism. This paper describes the multidimensional impacts of inequities for Indigenous tāngata whaikaha Māori (Māori with lived experience of disability). Semi-structured in-depth interviews were conducted with 28 tāngata whaikaha Māori and their whānau (extended family) using a kaupapa Māori Research methodology. An equity framework was used to analyse the data. The results describe: (1) inequitable access to the determinants of health and well-being; (2) inequitable access to and through health and disability care; (3) differential quality of health and disability care received; and (4) Indigenous Māori-driven solutions. These data confirm that tāngata whaikaha Māori in the nation-state known as New Zealand experience racism, ableism and disablism, compounded by the intersection between these types of discrimination. Recommendations from the data support the inclusion of tāngata whaikaha Māori in decision-making structures, including all policies and practices, along with equal partnership rights when it comes to designing health and disability systems and services.

## 1. Introduction

Approximately 15 percent of the world’s population experiences disability, and the prevalence is rising due to an aging population [1]. People with lived experience of disability have poorer health outcomes, lower education achievements, less economic participation, and higher rates of poverty [1]. Certain groups within this population experience even poorer outcomes because of the multiplicative impacts that occur from the many intersecting forms of oppression with ableism and disablism (for instance, colonialism, imperialism, racism, patriarchy, classism, xenophobia, homophobia, transphobia, religious discrimination). Although the terms, ableism and disablism, are both used here to describe disability discrimination, the emphasis for each of these is slightly different. Ableism is discrimination in favour of non-disabled people while disablism is discrimination against disabled people [2]. Indigenous peoples with lived experience of disability are one such example of a population experiencing poorer health outcomes due to these oppressions [3].

Intersectionality can be considered as an “experiential reality dynamically shaped by multiple, complex, intersecting, and interdependent systems, structures, and axes of power, privilege and oppression” [3] (p. 72) and is an important analytical approach to understanding the multiplicative impacts that occur for Indigenous peoples with lived experience of disability. Indigenous peoples worldwide have diverse historical and contemporary impacts of disablement arising from colonisation and coloniality, which are in themselves disabling [3,4]. Similarly, Indigenous peoples with lived experience of disability are more likely to experience the impacts of other forms of oppression, including racism [5,6]. Of significance, this group is largely invisible, especially with respect to the impacts of colonisation, coloniality and racism, disability legislation, and disability identity discourse [6,7,8].

In the nation-state currently known as New Zealand, Māori are tāngata whenua (Indigenous peoples) and have a unique partnership with the British Crown, represented by the New Zealand Government and legislated under Te Tiriti o Waitangi, the Māori version of the Treaty of Waitangi (Te Tiriti) [9]. While Te Tiriti guarantees existing tāngata whenua rights for Māori to self-determination and, amongst other things, to equity in all aspects, repeated cross-sectional survey data continue to demonstrate persistent inequities in health and socioeconomic outcomes over time [10]. This is due, in large measure, to repeated and ongoing breaches by the government in its obligations under Te Tiriti since 1840, which have resulted in the disenfranchisement of Māori from their lands, culture, traditional knowledge and practices, economic base, models of health and well-being, and collective governance and leadership [6,7,8]. While more recent government-funded initiatives led by kaupapa Māori (Māori-owned and governed) health and social services, and communities have challenged contributors to these inequities, there remains a scarcity of initiatives for tāngata whaikaha Māori [6]. Not only do the significant health and socioeconomic inequities for tāngata whaikaha Māori reflect a breach of Indigenous rights as set out under Te Tiriti, but also those rights, under international human rights instruments, such as (but not limited to) the: Convention on the Rights of Persons with Disabilities; Convention on the Rights of the Child; International Covenant on Economic, Social and Cultural Rights; Optional Protocol to the Convention against Torture and Other Cruel, Inhuman or Degrading Treatment or Punishment; and Declaration on the Rights of Indigenous Peoples [6].

Lack of definitional clarity around concepts of disability from the perspectives of Indigenous peoples, the absence of robust statistics and a paucity of disability research has resulted in limited evidence and gaps in all sectors across health and disability policy [4,6,11]. Although there is growing evidence regarding the relationships between disability, health and socioeconomic inequities, further evidence is needed into the exact nature of these relationships, their dynamics, and causalities [12]. The impacts are often nuanced and context-dependent, and although strong anecdotal evidence is available, there are few rigorous quantitative or qualitative evidence-based studies [12]. Even fewer data are available that provide evidence that is meaningful from the perspective of Indigenous peoples with lived experience of disability, or that recognise the intersecting multiplicative impacts of colonisation, coloniality and racism for these groups [3,4].

In this paper, we use the term ‘tāngata whaikaha Māori’ as an umbrella term for Disabled Māori or Māori with lived experience of disability. Tāngata whaikaha is a mana-enhancing (strengths-based) descriptor of people striving ‘to have ability’ or ‘to be enabled’. We acknowledge other terms are also used by Māori for impairment and disability-related concepts. We also recognise that internationally used terms such as ‘people with disabilities’ may be used however, adhering to the Social Model of Disability, where “*Disability is considered a result of society’s failure to meet the aspirations and needs of people. Disablement is thus imposed on people by society*” [13] (p. 1), we have chosen to privilege the terminological preferences of tāngata whaikaha Māori over Western disability terms.

The existing body of disability knowledge in New Zealand is primarily shaped without the specific views of tāngata whaikaha Māori, or their collective whānau (extended families) and communities. Considering that tāngata whaikaha Māori report the highest level of cultural, social, and economic deprivation in this country [14], the interrogation of the prominent ‘Western’ (i.e., colonially imposed biomedical) concepts of disability and impairment is essential.

The dominance of colonial knowledge and its imposition upon Indigenous nations is significant, evidenced by the notion of individual paramountcy that underpins health and disability policy [6,15]. The nature of collectivism among Indigenous peoples is therefore ignored. Furthermore, by claiming the status of ‘normal’ within a post-colonial society renders Indigenous groups ‘abnormal’, thereby using the concept of disability as a colonising tool [16]. Concepts of whakapapa guide Māori to incorporate historical and contemporary contexts of balancing all aspects of well-being, including the exploration of the ways in which Māori identify and refer to concepts of disability and impairment [6].

Colonial concepts of disability (i.e., individualized, Western, deficit-based), such as people ‘having disabilities’, have contributed to the existing challenges for tāngata whaikaha Māori regarding meaningful data through dominant medical and social models that are used to measure eligibility for services. While tāngata whaikaha Māori may benefit from being ‘visible’ to services, they may not choose to self-identify with the notion of impairment-based eligibility that remain in place to date [17]. In addition, measurements such as the ‘Washington Group Short Set on Functioning’ questions, currently used in the New Zealand Census as a proxy for ‘disability’ [18], are not culturally informed and do not actually measure disability (i.e., the process and effects of disablism on populations with impairment). Instead, this measure focuses on serving as an international and longitudinally repeatable measure of impairment that does not attempt to assess prevalence, or include measures that account for historical, political, social, cultural, and environmental contexts or impacts affecting Māori.

The resultant impacts for tāngata whaikaha Māori are wide-ranging. In the last New Zealand prevalence survey held in 2013, 26 percent of Māori, compared with 24 percent of the general population, self-reported disability [14,19]. However, when adjusted for the population age structure, the prevalence of tāngata whaikaha Māori is 32 percent [14]. Despite having higher rates of health-related impairment, 39 percent report an unmet health need (1.4 times higher than non-Māori) [14]. One in four tāngata whaikaha Māori report having insufficient income to meet their daily needs, and experience inequities in accessing funding for equipment and care [6,14]. Inequities in disability services for tāngata whaikaha Māori have been highlighted over the past decade with reports of insufficient assessments, treatment, and access to culturally acceptable support services [20].

Underpinned by kaupapa Māori theory and praxis [21], this qualitative study explores the multidimensional impacts of inequities for tāngata whaikaha Māori, and how these impact on holistic well-being. A specific objective of this paper is to provide recommendations for addressing any inequities impacting on tāngata whaikaha and their whānau.

## 2. Materials and Methods

A Kaupapa Māori Research (KMR) methodology was used throughout all phases of this research. The key to this methodology is the inclusion of a number of Māori principles that uphold the mana (respect, authority) and aspirations of the participants, and genuine engagement with the community as a partnership for research. Kaupapa Māori Research thus privileges the voices and perspectives of Māori and ensures that mātauranga Māori (Indigenous Māori knowledge) is not only acknowledged, but that it is prioritised [21]. Given the existing inequities across the health and disability system, evidenced internationally and locally for Indigenous peoples, we have drawn on an equity framework developed by Camara Jones to underpin our analysis and interpretations of this research [22]. This equity analysis comprises three main domains: (1) differential exposure to the determinants of health and well-being; (2) inequitable access to and through health and disability care; and (3) discriminatory/differential quality of health and disability care received.

### 2.1. Data Collection

A qualitative approach to data collection, incorporating flexible KMR methods, was initially planned to include marae hui (Māori group discussions at a meeting house) and wānanga (workshops), along with in-depth individual and whānau interviews. To align with traditional Māori ways of gathering information, these methods were largely planned to be in-person, with provision made for digital online interviews (for instance, via Zoom or phone) to provide various choices that would suit participants. Due to the COVID-19 pandemic public health restrictions, many of the public gatherings were not permitted during this phase so approximately half of the interviews were conducted online.

Semi-structured interview guides were developed with open-ended questions that explored topics within three broad domains: (1) culture and identity (including disability concepts); (2) health and disability services (including discrimination); and (3) transformation of the disability system (suggestions for improvements). Questions used included: Please tell us about how you identify yourself? What terms or concepts do you prefer to use yourself to describe disability? Have you ever been discriminated against when accessing health or disability services? What changes would you like to see immediately in the current disability system?

A strengths-based approach ensured all those involved with the project had control over who they shared their stories with, and how they wanted them to be interpreted and represented. One interview was conducted in New Zealand Sign Language (NZSL) assisted by an NZSL interpreter. Two interviews were conducted using te reo Māori (Māori language).

### 2.2. Participants and Sampling

A purposive sampling strategy was used to identify tāngata whaikaha Māori with lived experience of a range of impairments (for instance, mobility, sensory, cognitive learning, etc.) and their whānau across New Zealand. We aimed to recruit a minimum of 20 tāngata whaikaha Māori along with their whānau, parents or caregivers of children with lived experience of disability. Based on the teams’ previous KMR qualitative studies, we estimated this would ensure a wide diversity of views and perspectives about culture and identity, and importantly, it would provide data saturation. The inclusion criteria were tāngata whaikaha Māori (≥18 years) who self-identified with lived experience of disability and who were able to consent to the research.

### 2.3. Recruitment

In practical terms, the KMR methodology described above was enacted throughout the research process by using the Whānau Tuatahi (Family First) framework which was developed from a Māori community research partnership. The principles of the Whānau Tuatahi framework applied are whakawhirinaki (trust), whakawhanaungatanga (relationship-building), whakamana (empowerment), ngāwari (flexibility), utu (reciprocity) and hurihuringa (reflexivity), and are reflected in the application of the methods described [23].

Following advice from a range of Māori disability groups, the participant information and consent forms, and letters of invitation were written in English using a health literacy approach. They were then translated into te reo Māori and accessible information formats, such as, ‘Easy Read’, by expert translators. A letter of invitation was sent using existing Māori disability networks and social media. A snowballing technique was used to include a range of geographical regions, ages, genders, and impairments. A full explanation of the study, including how we would de-identify and use their data, was provided to all participants prior to data collection, and opportunities were given to everyone to consult and involve whānau and ask questions prior to informed consent being obtained from those willing to participate.

Participants were offered the language of their choice for the interviews. Most interviews were conducted in English, although in many cases, te reo Māori was used with English. For two interviews, te reo Māori was used for the interview and these were conducted by a fluent te reo Māori interviewer. Reasonable accommodations and supports were offered to suit each participant’s needs, such as the offer of accessible information and provision of NZSL interpreters. To guarantee a high degree of fidelity, all researchers were trained and mentored by the principal investigator (BJ) to ensure the data being obtained were coherent with the research question.

Māori disability researchers, including tāngata whaikaha Māori researchers, collected all data. Interviews were conducted in person or by Zoom, and digitally recorded, then transcribed and checked for accuracy. Transcripts were de-identified and assigned a study number before distributing them to the research team for thematic coding.

### 2.4. Ethical Considerations

Research ethical approval was obtained prior to the commencement of the study by the Southern Health and Disability Ethics Committee (Reference: 19/STH/153).

An expert Māori community steering group was established to provide oversight throughout the research process. This group consisted of 15 tāngata whaikaha Māori and their whānau (extended family) with a wide range of disability expertise.

‘Te Ara Tika Guidelines for Māori Research Ethics’ were used to develop and conduct all aspects of this research process [24]. As researchers, we adhered to KMR principles throughout all stages of the project. We collectively ensured participation was mana-enhancing (especially for tāngata whaikaha Māori), followed tikanga, and benefited the people and communities who have been involved in this research.

### 2.5. Analysis

Kaupapa Māori Research principles, as outlined in the methodology section above, has informed all stages of this research, including the application of an equity framework that enabled the analysis process and interpretation of the data [22]. This equity framework was used to organise the deductive thematic domains. For each of these three domains, the intersectionality between racism, ableism and disablism was inductively explored.

All transcripts were initially coded into sub-themes by the researchers undertaking the interviews and then coded by a second researcher. Participants were given the opportunity to review their transcripts before analysis, and comment or make changes. Emerging themes were then generated from the sub-themes and agreed on by two Māori researchers. Finally, to ensure analytic rigor and expert checking, the analysis was discussed via wānanga (the process of knowledge creation) with senior Māori researchers, including tāngata whaikaha Māori, who provided an independent Māori lens to the interpretation. This was an iterative process over several months, which included using a uniquely Indigenous Māori perspective on disability that is holistic and based on spiritual, collective, and relational values [21,25].

All researchers discussed their own positionality, including as either tāngata whaikaha Māori or whānau, reflecting on any possible biases or influence on the research. The research team combined more than 30 years of working with people to reduce racism, disablism and the intersectional inequities that arise from these. Reflexivity involved a process of whakawhiti kōrero (debate and negotiation) which has been described as an active and inclusive discussion that whakapapas back to traditional Māori ways of sense-making and negotiating solutions [26]. This was led by senior Māori researchers, including tāngata whaikaha Māori, ensuring Māori protocols were followed during this process. The draft themes and statements were read aloud, and the semi-structured, guided questions were used to help structure the negotiation around each theme until an agreement was reached.

To ensure data triangulation, the following processes were included. The collection of data was conducted through multiple methods, including interviews, focus groups and hui across New Zealand. Theory triangulation was incorporated by using multi-professional perspectives of KMR paradigms, an equity framework [22] and human rights disability theory [7]. To ensure data sovereignty [27], draft themes and interpretations were also shared with participants for feedback, commentary or editing.

## 3. Results

We interviewed a total of 28 tāngata whaikaha Māori participants with lived experience of disability, including three parents/whānau of tamariki (children) with lived experience of disability, from across rural and urban New Zealand. Aligning with participant preferences and ethical requirements to maintain anonymity, only generic demographic information is included here while more specific demographic details have not been disclosed for publication. Participant impairment types included: physical (*n* = 10), sensory (deaf, blind) (*n* = 5), neurodiverse (*n* = 6), learning/intellectual (*n* = 7) and mental health conditions (*n* = 7).

Of the 28 participants between the ages of 20 and 69 years, 9 were aged 20–39 years, 15 were 40–59 years and 4 were 60–69 years. Twenty-seven participants identified their ethnicity as Māori, with a wide range of iwi (tribal) affiliations, and one whānau identified as European New Zealander. Twelve participants were male (tāne) and fourteen were female (wāhine), one referred to being ‘atua’ (God-like) instead of a binary gender option and one did not answer. In alignment with our focus on holistic well-being, quotes were assigned pseudonyms relating to plants that are used for rongoā Māori (Māori healing and well-being).

The first three themes are presented within the equity framework discussed in the methodology section: (1) inequitable access to the determinants of health and well-being; (2) inequitable access to and through health and disability care; and (3) differential quality of health and disability care received. The fourth theme of re-Indigenising the system, outlines proposed Indigenous Māori-driven solutions. Additionally, experiences of racism, ableism and disablism, and the intersection between these types of discrimination are evident across each of the themes.

### 3.1. Inequitable Access to the Determinants of Health and Well-Being

Education, training, and employment were commonly reported by participants as important determinants of health and well-being for tāngata whaikaha Māori. Participants with lived experience of learning impairments shared their stories of how they were judged for being slow learners. Instead of being supported, they described feeling daily discrimination within both the education and health systems:


*“We come from the other side of the tracks where we would be looked down on; and the kids would not be doing well at school, and you’ve got inherent prejudices from the teachers as well as when you go into the hospital. You would get looked down on, and maybe it’s because they… just don’t like us. Like people who appear and look uneducated or not smart.” Kowhai*


It was also acknowledged that the education system does not have capabilities or processes to adequately identify or monitor whether it is performing adequately for Māori with lived experience of disability, including for tāngata turi (Māori deaf):


*“I think New Zealand wasn’t good enough at providing resources to Māori deaf children …or older Māori deaf people. Where is the information about them? Where did they go to school? What happened to those kids?” Kumarahou*


The impact of being brought up as a child under government-funded institutional care was highlighted as not only affecting participants, but also generating intergenerational effects:


*“How do I feel comfortable explaining my childhood? What I’ve discovered is for me to feel comfortable about that I need to acknowledge it. When my children have grandchildren, they’re going to be talking about their whakapapa [ancestry] as Māori, disability and me being in the Child Youth and Family system.” Manuka*


For Māori, traditional links to their whakapapa (ancestry), whenua (the land, rivers, and mountains) and tikanga (traditional culture and protocols) are expressed as important elements of their culture and identity. Participants reflected that they had retained a sense of primary self-identity as Māori, and their lived experience of disability had in many cases undermined their practical exposure to their own culture and compromised their ability to draw on their cultural connections within te ao Māori, which can compound disability or even be disabling in its own ways:


*“As tāngata Whaikaha Māori, I need to connect firstly as Māori as well as tāngata Whaikaha. My disablement is dual. I feel …inadequate in the sense of not being Māori enough, as well as being disabled... having had more of a Pākehā [non-Māori] upbringing and not having the whakapapa links … makes it challenging for me.” Nikau*


It was stated clearly that despite being classified as disabled from a Western persona, for Māori, this was not culturally appropriate:


*“My thought is that having a disability doesn’t define who I am. From a Western worldview they try and categorise us. They put us in that one box and go, “You’re disabled.” But from a te ao Māori [Māori worldview] perspective it’s who we are, and we walk with that journey, and we continue to embrace our te ao Māori.” Koromiko*


The opportunity to realise one’s cultural potential is an important contributor to cultural well-being. When disability impacted their opportunities, participants described lifelong barriers that not only limited their physical access to society but also contributed to their feeling of disconnection from their culture:


*“For Māori, the whenua [land] is a huge part of everything. Papatūānuku [earth mother] is it. I think being disabled in that manner has hindered me from having full identity, or be able to go right down that river, or right up that maunga [mountain], or down by the ocean.” Kawakawa*


One participant specifically noted that their learning disability, combined with limited connection to their Māori whānau and iwi (tribe), meant that learning te reo Māori was difficult:


*“I can’t speak Māori so I can’t do that, but I’m Ngāi Tahu [tribal name]. I find it challenging because I’ve got a learning disability. I’ve tried to go to courses that teach you Māori, but I haven’t been able to really understand it completely.” Makomako*


Other participants talked about how the limited access to marae (traditional Māori meeting places) impacted them in terms of feeling excluded:


*“My wife’s main marae is fully accessible, well in the wharekai at least. Wharenui [meeting house] isn’t [accessible] but that’s a flood area so they build things up on, you know, big steps and foundations. That’s when you notice you’re different again.” Nikau*


Discrimination was experienced as a result of disabled children being systematically removed from their whānau. One participant tried to make sense of this by highlighting the paternalism of the State. She discussed the justification for removing babies from their parents under forced adoptions:


*“Our disabled children got taken away, do you think this whole concept of disability for those that were born with it was about taking us away? They thought that we would be better off in institutions, or with Pākehā families than with our own people…because, as Māori, they didn’t see that we were capable of looking after our own.” Ponga*


### 3.2. Inequitable Access to and through Health and Disability Care

Tāngata whaikaha Māori, as well as being Māori, have additional needs that require the health system to be not only culturally safe but also non-disabling and accessible:


*“[The term] Māori disabled can be helpful, because see we still have to push. Because we’ve been discriminated against, unfortunately, we have to put ‘Māori disabled’ to remind people that we do have Māori disabled but put it in the nicest possible way.” Rewarewa*


The need for access to health services that provide cultural, holistic connections to te ao Māori, was seen as essential to maintaining hauora (well-being) for tāngata whaikaha. This included, for example, having personal access to traditional Māori medicines (Rongoa Māori):


*“I’ve also got lots of Rongoa [Māori medicine] and my uncle was a [tohunga] healer. I would always go there every day.” Titoki*


This need for kaupapa Māori services was further highlighted by one whānau who tried to install cultural content, within a sterile hospital environment, using visual representations of the whenua, the marae and iwi/whānau connections to support the person’s recovery:


*“So, there was a whole wall of life that surrounded him. So, I went like this every day, and I slapped up pictures or photographs or anything relevant [to his culture]. That was the maunga, the marae, the awa, the haerenga [mountains, meeting place, river, journey]... So, I just rammed it around all the walls.” Totara*


As well as no access to holistic cultural service, some participants explained how they also had to adapt their identity to a colonial, deficit-based paradigm in order to access services:


*“I use the word disability a lot because it’s a word that ableists can relate to. I’m not ashamed of it. I promote disability by being disabled, I’m proud of what I have achieved [despite] my disabilities, but it’s a word that [non-Māori] people know.” Rewarewa*


Even if participants adapted their identity to conform to the parameters of the service, as everyone in society does, the current health and disability systems imposed additional and unrealistic expectations on them. They discussed navigating complicated, institutional pathways to access funding unsuccessfully, compelling participants and whānau to provide substantial, unfunded support for their needs to be met:


*“Natural [unpaid] support is what ACC [accident insurance scheme] makes reference to… So, there’s a huge expectation that whānau have to cover the shortfall.” Totara*


The consequences of poor access and unmet disability needs were stated in relation to discrimination on the basis of disability. This discrimination was explained as a contributor to fewer life opportunities, resulting in negative impacts throughout the life course:


*“…all the missed opportunities, and how I treated our marriage… I feel like a bit sad and bitter.” Rata*


### 3.3. Differential Quality of Health and Disability Care Received

Participants spoke of generations of living with racism and how Māori had learned to adapt to Western culture in order to survive. For many, this meant engaging with a Western worldview while holding on to te ao Māori (a Māori worldview). One participant explained that:


*“Before colonisation there was no term… we were all one.” Post colonisation generations of Māori were forced to adapt and now “we’re able to walk in both worldviews.” Koromiko*


Another spoke despairingly of disablism, stating:


*“I’m sick of being discriminated against; being in a wheelchair that’s a problem for people.” Toetoe*


The effects of engaging with both a racist and ableist health and disability system were emphasised by a wāhine when she described how disabled she became when using Pākehā services:


*“Well actually the only time I become fricking disabled is when I’m accessing Pākehā services, you know, or the environment.” Ponga*


Interpersonal and institutional racism, compounded by ableist attitudes, were experienced by most participants receiving low-quality health and disability services. For one participant, being outspoken and fighting for her rights in a system that does not meet her needs was interpreted by a service provider as her being violent and mentally unstable, and resulted in a breakdown of the therapeutic relationship:


*“[They treat us] like, ‘Dumb Māori!’ or ‘Oh disabled, oh look out for her!’ and that we’re always violent; Māori women, disabled women, because we are passionate about fairness and validity, and excellence and education. We’re seen as out of control, mad and unstable.” Rewarewa*


Some health professionals reportedly expressed a clinical view that disability was a deficit that would burden the parents during the life course of their disabled child. Two Māori parents discussed how they were repeatedly told to abort their unborn baby who would be born with an impairment rather than providing options for progressing with the pregnancy and available supports:


*“We went so many times… for scans, and every time we went a new doctor would offer me an abortion, and I was so angry. I remember yelling at them, … it was almost offensive that there was an expectation.” Rimurapa/Rimuroa*


The lack of understanding from a te ao Māori perspective was experienced as insulting and eugenic by the family who, in contrast, spoke of ‘an underpinning acceptance’ that all people are born with an innate uniqueness. They shared their strengths-based perspective, describing their unborn baby as *“a gift to the whānau and something amazing”* and how they just had to wait and see what that gift would be:


*“I think that comes from our views that children have these unique characteristics, and we emphasize their strengths, and we support them to reach their possible outcomes, and I think disability is just part and parcel of that person but doesn’t define what an amazing contribution they will have.” Rimurapa/Rimuroa*


The need to “*find people that share the same values as you*” was highlighted as a critical part of meeting tāngata whaikaha needs. However, a lack of being able to access kaupapa Māori services was seen as a barrier, when the only choice of disability services was non-Māori:


*“[Disability] providers, they run on the Pākehā, white, Western culture; not te ao Māori.“ Koromiko*


The medicalised process of assessing the needs of disabled Māori was experienced as culturally inappropriate by participants, who commented that, “*the process and the content, listing unrelated questions for Māori just doesn’t work*” (*Ponga*). Some said they “*felt like a stunned mullet*” during the assessment process, with others unsure of the whole process. The importance of having a disability Māori workforce was highlighted, with the suggestion that disability needs assessors must be part of the service, along with a culturally safe space or environment while being assessed:


*“I would think that the assessor should be Māori, and the environment that it’s [the assessment] is held in would be conducive to [disabled Māori], what they are comfortable with.” Kiekie*


A lack of health literacy approaches, used by health professionals discussing disability services, was also emphasized by participants:


*“No one told us our [disability] entitlements, we would just go through [the whole process] without knowing what we are entitled to.” Toetoe*


The power imbalance within the health and disability system was explained by participants when discussing the unfairness of the system towards people with certain types of impairments at the exclusion of others:


*“I also think that there is a lot of biases in the [health and disability] sector and biases towards a certain type of people who have a disability. When I think of the disability system, I think of who is in that echelon of influence.” Ponga*


A lack of accountability within disability services was discussed as another contributor to poor quality health and disability services:


*“Lack of accountability as well, I think that that’s really all wrong. Māori who have reached out for disability help but not got it, and then they have [negative] consequences. They [health professionals] should be held accountable.” Ponga*


These results demonstrate the multiplicative impacts of racism, ableism and disablism in New Zealand across exposure to determinants of health, access to health care and the quality of care received. Despite these experiences, participants willingly proposed solutions that could help address these inequities.

### 3.4. Re-Indigenise the System

One participant discussed the benefits of a kaupapa Māori lens for transforming the entire health and disability system rather than fitting Māori into the current system. This included a desire for holistic health services, such as traditional Māori methods of healing and Māori knowledge of well-being, alongside clinical medicine:


*“… a lot of it is we’re not accessing our traditional medicines and knowledge as well. It would be good to have maybe a set-up of the two; if we were to have rongoā Māori or Māori medicine and treatment in each of our hospital systems we would actually see Māori thrive better.” Rata*


As well as providing choices and access to traditional medicines, it was proposed that the whole system needed to be completely re-Indigenised rather than continually trying to decolonise it:


*“…re-indigenise the health and disability system, not de-colonise, but to re-indigenise it. Only because Pākehā, or the white man, has had all these years, you know 150 plus years of doing it their way and it hasn’t worked for our people.” Piripiri*


The issue of power was highlighted by one participant saying the control was held by able-bodied people and excluded disabled people from decision-making when it came to developing policies that directly impacted on their lives. They stressed the importance of having their voices heard within disability services and policies:


*“They [health and disability policy makers] don’t include us in a lot of their thinking. They don’t. What upsets me is that they have all these able-bodied people making decisions for those of us who are disabled, thinking that they know best.” Koromiko*


The plea to not focus on the ableist ‘deficit’ perspective, but instead to be respected as valued members of society was a clear message from all participants:


*“I wish people would just accept us for what we can bring to the table for a lot of situations that are impacting this country right now.” Rewarewa*


Participants also emphasised the benefits of the inclusion of disabled Māori exercising their self-determinism and contributing to society as a whole:


*“I would like them to know that we have … a lot of talents. If we can’t move, we’ve got creative thinking; we’re abstract thinkers. All we want to do is to contribute like abled people; to contribute to society, to contribute something to the economy, to contribute something to mental health and disability, and not be attacked.” Rewarewa*


## 4. Discussion

There are clear and pervasive inequities in experiences and outcomes for tāngata whaikaha Māori in New Zealand. These data, while not intended to be generalisable, further confirm that tāngata whaikaha Māori are experiencing racism, ableism and disablism. These discriminatory practices are resulting in multiplicative biases (intersectionality), which are impacting negatively on tāngata whaikaha Māori, whānau and societal well-being. While the data were thematically analysed a priori based on an equity framework, there were clear and cross-cutting issues of racism and disablism that emerged. The same biases, discriminations and ableism are evident through the differential exposure to social determinants of health, access to appropriate health care, and the quality of care. Consequently, multidimensional impacts are experienced throughout the life course. These impacts are not linear, they are cyclical, and reinforce a poorer quality of health care experience which negatively impacts on future life opportunities.

### 4.1. Intersectionality

We have previously highlighted intersectionality as “an essential analytical tool and critical vantage point” [3] (p. 80) with regard to understanding the drivers of significant inequities for Indigenous peoples with lived experience of disability. Primary issues hindering the expansion of disability studies for Indigenous peoples have been linked to the lack of empirical studies [28] and the failure to explore the experiences of Indigenous peoples with lived experience of disability [29]. Hickey and Wilson describe the cumulative disadvantage of being Māori and having lived experience of disability, and the diverse impacts of disablement for Indigenous peoples arising from colonisation, racism, and dysfunction that are in themselves disabling. They emphasise the disproportionately high rate of disability in Indigenous communities compared with non-Indigenous peers and the gaps in evidence for well-being [8].

Our findings support this critique and demonstrate the negative effect that the loss, removal, and prohibition of culture have on Indigenous Māori. However, the intersections of Indigeneity and disability may not only create summative disadvantages; there may be a potential benefit. For example, being Māori might mitigate the disadvantage of disability because of its strength-based perceptions of disability, level of social responsibility and collectivism culture. This has not been explored to date. The current systemic discrimination that exists in New Zealand does not allow for alternate views of disability to be sustained within the dominant society.

The colonisation of New Zealand brought with it different concepts and institutions that aggressively privileged Western ontologies and epistemologies. One of the many impacts of colonisation, highlighted by some participants, was the need for ‘white superiority’ over Indigenous Māori, including tāngata whaikaha Māori. One of the results of this enduring power imbalance has meant that tāngata whaikaha Māori were, and continue to be, subjected to the superimposed values of disability from the colonising culture—a culture that largely ignores the holistic nature of Indigenous well-being [6].

The divergence of cultural paradigms around ‘disability’ is particularly striking for tāngata whaikaha Māori who are interrogated using deficit, non-Indigenous health assessments that diverge from Māori worldviews. There has been some work highlighting the limitations of existing mainstream outcome frameworks and advocating for a principled approach to outcome determination (including the need for strengths-based and Māori-focused outcomes) [30,31]. Culturally appropriate assessment tools perceive tāngata whaikaha as valuable members of the whānau and community rather than a burden or ‘nuisance’, resulting in fit-for-purpose outcome measurements [32].

### 4.2. A Right to Equitable Outcomes and Access to Health Care

Tāngata whaikaha Māori experience many differences in life opportunities, with for instance, fewer job opportunities, differential access to their culture and community, and increased stresses that result in differences in their underlying health status [6]. Our results indicate that factors, such as negative attitudes of superiority of the system knowing what is best, the imposition of formulaic services, and differences in opportunities, expectations, and the perceived ability to self-manage, start early and continue across the tāngata whaikaha Māori lifespan. Participants recounted paternalistic presumptions made by people who had more power both in the health and other systems. The removal of rangatiratanga (self-determination and autonomy) across these systems is an effective means of subjugating an Indigenous population and undermining their health, and is a violation of Te Tiriti.

Differences in access to health and disability care are experienced by tāngata whaikaha Māori. These include delayed access to preventative services and lesser referrals to curative treatments. Accessing disability services relies firstly on tāngata whaikaha Māori identifying or labelling themselves as ‘disabled’. This is at odds with Māori, who first and foremost identify with te ao Māori, with the desire to maintain their cultural identity being central to hauora (well-being) [33].

People who receive less care in the health system, experience an increased residual burden which means more exposure to the impacts of social determinants and consequently an ever-increasing, self-reinforcing, negative impact on well-being in a dwindling spiral. The better the health system is at mitigating the impacts of impairment and the sooner it can recognise that special measures are required, which necessitate additional resources (people and funding) to address the added prevalence and impact of impairment, the greater the benefit to society as a whole.

In New Zealand, Te Tiriti requires the Government to “commit to achieving equitable health outcomes for Māori” [34] (p. 163). New Zealand’s Waitangi Tribunal has found that this obligation is heightened when, as is the case for tāngata whaikaha Māori, the Government also knows about the ongoing impacts of colonisation and racism, and the enduring impacts they have on poor health outcomes [34]. However, our results indicate that health and disability systems in New Zealand continue to work against principles of rangatiratanga, options, equity, and active protection, along with failing to meet the human rights expectations of transparency of and participation in decision-making.

Article 26 of the Convention on the Rights of Persons with Disabilities explicitly states that people with lived experience of disability have a right to habilitation and rehabilitation to maintain their health and well-being. This requires proactive investment in maintaining well-being until a return to function and participation, as defined by the person, is achieved. Participation in this context is wider than achieving basic personal tasks in the home; it also encompasses social connection with their whānau, the community and their culture [35].

### 4.3. Kaupapa Māori Services and Access to High Quality Care

Our results show that tāngata whaikaha Māori and their whānau experience culturally unsafe care in the current New Zealand health and disability system. Tāngata whaikaha Māori require better access to quality services and culturally competent services, delivered by staff with cultural awareness and competency. The current view of health and disability is biologically based and predominantly focused on the individual. To understand an Indigenous perspective, health systems, organisations and staff need to think more broadly about the collective whanau and community environment along with other determinants of health.

For tāngata whaikaha Māori, access to culturally appropriate disability services is extremely limited, with only 33 of 980 Ministry of Health-funded disability providers identifying as Māori-owned and Māori-governed [6,34]. These low numbers severely limit the ability for tāngata whaikaha Māori to access disability support services from providers with Māori worldviews, underpinned by cultural values, utilising Māori models of well-being [34]. Our results indicate that the lack of access to kaupapa Māori providers creates a barrier to accessing levels of care and disability support that meet their needs, build connections (to communities, to whānau and to culture) and support their aspirations. While there are greater numbers of Māori health providers available, there remain challenges to access to these for tāngata whaikaha Māori. Additionally, there have been a lack of government policy support and decades of underfunding of Māori health providers, along with government contracting practices that are based on siloed funding models and individualistic and reactive systems developed without proper input from Māori. These place a high burden of audit and reporting requirements on Māori providers which do not align with their principles of holistic care [34].

### 4.4. Mitigation Strategies

While building the range of services offered by Māori providers is an important step towards creating health and disability systems that support tāngata whaikaha Māori well-being, there are still obligations on all publicly funded health and disability services to operate in a way that is culturally safe, which have been reinforced by the recent passing of the Pae Ora (Healthy Futures) Act 2022 [36]. Regulated health professionals also need to meet standards of cultural competence under the Health Practitioners Competence Assurance Act 2003 [37]. Some progress is being made in improving accountability, with public and private health and disability services also having to now meet the requirements of the ‘Ngā Paerewa Health and Disability Service Standards’, which include a new set of more robust cultural safety and service-user engagement requirements [38].

Currently, tāngata whaikaha Māori are not identified as a priority population within the disability services and many have no access to the supports they require to live their lives, especially in a culturally meaningful way. The impacts of multiple intersecting forms of oppression such as racism, ableism and disablism are not currently analysed, and all sectors need to research this. Māori disability data are a living tāonga (treasure), and measures should be developed and governed by Indigenous peoples to advance the aspirations of tāngata whaikaha Māori [27].

Health and well-being do not start and end with the health system. All the other systems and environments influence well-being, and therefore a critical role of the health and disability systems is to work outwards in recognizing the effects of these on health outcomes. The health system is a microcosm of society and the staff within this system are a reflection of societal attitudes, so there is a need to proactively address these attitudes within health educational institutions and with employers of the health care workforce. We need to proactively address racism and disablism in the training of health staff, but also need a diverse workforce that proactively includes those that are disadvantaged by the systems so that more diverse attitudes can start to shape and shift the normal values of the workforce [39].

Mitigation strategies firstly need to address the social determinants of equity which demand an ‘all of government’ authentic partnership approach [22]. Focusing on the social determinants of health provides a platform from which disadvantaged people can be valued equally. While in New Zealand, the newly established Whaikaha: Ministry of Disabled People has a critical role to play regarding redress, and ensuring tāngata whaikaha Māori are visible and their voices are heard and enacted, which lies with all government sectors. Recognising historical injustices is a vital step to eliminating inequities and the current Waitangi Tribunal hearings into claims by tāngata whaikaha Māori are gathering evidence to support this [40]. Suitable compensation will be needed for tāngata whaikaha Māori and finally resourcing this group according to need is essential. Monitoring for inequities across all health and social sectors is key to measuring and preventing inequities, including an equity criterion that allows for the analysis to accurately identify who is experiencing disadvantages and then mitigating them.

In summary, this paper describes novel data insights directly from tāngata whaikaha Māori which highlight the challenges and provide solutions that are the priorities of this population. While there is a body of evidence for disability research, and the links between racism and negative health outcomes [41], there is a paucity of Indigenous-led disability research exploring the intersections of both types of discrimination. Our findings align with the literature that shows exposure to multiple forms of discrimination is associated with poorer health in New Zealand [42]. Embedded in the Māori identity of disabled persons and their whānau, this research draws on mātauranga Māori (Māori knowledge), and seeks to encompass Indigenous perspectives of disability and the uniqueness of all Māori. Our findings support a mandate for a systemic change for tāngata whaikaha Māori with a whānau approach that prioritises and resources those with the most needs, enabling the fulfilment of their cultural values and participation in their own communities.

### 4.5. Strengths and Limitations

A strength of this research was that it was designed, led, and conducted by a predominately Māori team, including senior researchers with decades of lived experience of disability. All the participants involved were tāngata whaikaha Māori and their whānau with lived experience of disability. This process has contributed to the validation and creation of Indigenous Mātauranga Māori knowledge from a disability perspective and provides rich insights not only for Māori, but insights that might inform disabled Indigenous peoples worldwide.

## 5. Conclusions

This paper has described the multidimensional impacts of being both Māori and having lived experience of disability in New Zealand. It is critical that the intersections of racism, ableism and disablism are identified, monitored, and eliminated to avoid the harmful effects. Urgent action is needed to eradicate discrimination towards tāngata whaikaha Māori, in line with the obligations under the Te Tiriti, the Convention on the Rights of Persons with Disabilities, the Declaration on the Rights of Indigenous Peoples and other international human rights instruments, to allow for their cultural rights to connect to te ao Māori (the Māori world), and to live free from all forms of discrimination. Interventions are needed that include tāngata whaikaha Māori in decision-making structures, comprising all policies and practices, and equal partnership rights when it comes to designing systems and services that impact their lives. To enact change, there is a need to urgently set up proactive, multiagency collaborations using an intergovernmental, health and disability-in-all-policies approach.

## Data Availability

The data presented in this study may be available upon request from the corresponding author. The data are not publicly available due to Māori data sovereignty principles.

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
