# Peer review of "The Multidimensional Impacts of Inequities for Tāngata Whaikaha Māori (Indigenous Māori with Lived Experience of Disability) in Aotearoa, New Zealand"

_ijerph, 2022, doi:10.3390/ijerph192013558_

Round 1

Reviewer 1 Report

Thank you for this important article presenting the findings of what reads as a robust research project. It is very well written and clearly illustrates the findings with important centrality of the participants' voices through the quotes. It is also strongly positioned within the health and disability literature and current statistics.

There are minor points to consider:

1. Introducing the concepts of ableism and disablism earlier in the article - where these concepts are first referred to (p1 lines 42 - 44) for example

2. Some further discussion about the inadequacy of the 'Washington Group Short Set on Functioning questions' used in current NZ statistics is needed. The explanation about this inadequacy is not clear to me. I recognise that is it not a culturally informed set of questions and wonder if this is the point being made?

3. Later in the article it is made clearer how the Māori disabled advisory group was engaged with in the research, informing and reflecting the Kuapapa Māori methodology. In the Data Collection section where the development and focus of the semi-structured interviews are referred to (p 4 lines 160 -166) it would be useful if some discussion of how the co-development of interview questions was undertaken  - also some short examples of the questions would be illustrative.

4. Further information about the lived experience of disability of the 25 (of the 28) disabled participants would also be useful  - the range of experiences of disability are referred to but there is no breakdown of how many people for example were learning disabled, deaf etc. I understand this may not be the most important aspect of their experiences they are reporting in the research but it would be useful. Likewise some indication of intersecting experiences in addition to culture/race eg sexuality/ies; age; rural/urban living.

5. Translation of Māori words and terms is very helpful - check the manuscript to ensure this is done throughout - there are some places where translation is not included in (  ) after the Māori term.

An excellent article that contributes new, timely and important knowledge.

Thank you

Author Response

Thank you for the review - our responses are enclosed in the attachment.

Reviewer 2 Report

Thank you for the opportunity to review this article and your work. The article is very well-written and accessible to the reader. The literature presents a clear case for the need to conduct the study, and also the relevance of the methodology. The methodology itself is both warranted and of interest to readers. The study provides insight into the particular experiences of disabled Maori people, the importance of their cultural knowledge and an in-depth understanding of their concerns related to accessing services. This study is of interest to Indigenous peoples globally, and researchers who are also engaging in the work of understanding disability from the standpoint of those who see human difference as a gift.

I recommend the article to be published in its current form.

Author Response

(The authors gave the same response as above.)

Reviewer 3 Report

Very interesting article. Term "lived experience with a disability" is very unfamiliar to me. Can you just use "with disabilities"?  That is what is in the literature. Also, use people first language -- Maori with disabilities versus disabled Maori.  What were the questions asked during data collection? What was the make-up pf the sample population? This section needs to be more specific (i.e., gender, specific numbers of participants and disability, etc.). Editing throughout and check noun-verb agreements.

Author Response

(The authors gave the same response as above.)

Round 2

Reviewer 3 Report

Authors, thank you for responding to my queries and addressing the appropriate concerns.